# A Systematic Review of the Relationship between Chest CT Severity Score and Laboratory Findings and Clinical Parameters in COVID-19 Pneumonia

**DOI:** 10.3390/diagnostics13132223

**Published:** 2023-06-29

**Authors:** Naif A. Majrashi, Rakan A. Alhulaibi, Ibrahim H. Nammazi, Mohammed H. Alqasi, Ali S. Alyami, Wael A. Ageeli, Nouf H. Abuhadi, Ali A. Kharizy, Abdu M. Khormi, Mohammed G. Ghazwani, Ali A. Alqasmi, Turkey A. Refaee

**Affiliations:** Diagnostic Radiography Technology (DRT) Department, Faculty of Applied Medical Sciences, Jazan University, Jazan 45142, Saudi Arabia; alhulaibi.rakan@gmail.com (R.A.A.); ibrnma88@gmail.com (I.H.N.); moh.hurubi@outlook.sa (M.H.A.); aalmansour@jazanu.edu.sa (A.S.A.); wageeli@jazanu.edu.sa (W.A.A.); nabuhadi@jazanu.edu.sa (N.H.A.); akharizy@jazanu.edu.sa (A.A.K.); akhormy@jazanu.edu.sa (A.M.K.); mghazwni@jazanu.edu.sa (M.G.G.); abualwah2010@hotmail.com (A.A.A.); trefaee@jazanu.edu.sa (T.A.R.)

**Keywords:** systematic review, CT, severity score, COVID-19, laboratory findings, clinical outcomes

## Abstract

The COVID-19 virus has infected millions of people and became a global pandemic in 2020. The efficacy of laboratory and clinical parameters in the diagnosis and monitoring of COVID-19 has been established. The CT scan has been identified as a crucial tool in the prognostication of COVID-19 pneumonia. Moreover, it has been proposed that the CT severity score can be utilized for the diagnosis and prognostication of COVID-19 disease severity and exhibits a correlation with laboratory findings such as inflammatory markers, blood glucose levels, and clinical parameters such as endotracheal intubation, oxygen saturation, mortality, and hospital admissions. Nevertheless, the correlation between the CT severity score and clinical or laboratory parameters has not been firmly established. The objective of this study is to provide a comprehensive review of the aforementioned association. This review used a systematic approach to collate and assess the existing literature that investigates the correlation between CT severity score and laboratory and clinical parameters. The search was conducted using Embase Ovid, MEDLINE Ovid, and PubMed databases, covering the period from inception to 20 May 2023. This review identified 20 studies involving more than 8000 participants of varying designs. The findings showed that the CT severity score is positively associated with laboratory and clinical parameters in COVID-19 patients. The findings indicate that the CT severity score exhibits a satisfactory level of prognostic accuracy in predicting mortality among patients with COVID-19.

## 1. Introduction

The emergence of the novel coronavirus, which causes coronavirus disease 2019 (COVID-19), was initially reported in China and has since disseminated worldwide with remarkable speed, affecting a significant population of individuals since the onset of 2020 [1,2]. The etiology of COVID-19 can be attributed to the novel Severe Acute Respiratory Syndrome Coronavirus-2 (SARS-CoV-2), a member of the coronavirus family. Previous studies have shown that SARS-CoV-2 shares up to 70% amino acid identity and structural similarity with other coronaviruses [2,3,4]. It has been reported that a cluster of pneumonia cases exhibiting severe respiratory symptoms and clinical manifestations were attributed to SARS-CoV-2 as the causative agent [5]. The prevailing clinical manifestation of SARS-CoV-2 typically comprises symptoms such as fever, cough, dry cough, dyspnea, headache, dizziness, generalized weakness, vomiting, diarrhea, and respiratory tract symptoms. The clinical manifestation that has been observed to be of utmost significance in COVID-19 patients is the respiratory tract system, although COVID-19 has been demonstrated to impact other bodily systems as well [5]. The diagnosis of COVID-19 pneumonia is known to be based on clinical symptoms, laboratory findings, and imaging studies.

SARS-CoV-2 exhibits a wider range of community transmission in comparison to other coronaviruses such as severe acute respiratory syndrome (SARS-1) and Middle East Respiratory Syndrome (MERS), which also belong to the same coronavirus genus [6]. Consequently, laboratory analysis could potentially assume a crucial function in differentiating COVID-19 from alternative respiratory ailments. L aboratory testing parameters have been observed to be effective in managing the COVID-19 pandemic, monitoring comorbidities, identifying complications, choosing appropriate treatment methods, and evaluating the incidence of diseases in the population [6]. The variability in laboratory testing strategies across countries is contingent upon the accessibility of diagnostic techniques and materials. Research has indicated that COVID-19 patients who were diagnosed with RT-PCR exhibited elevated levels of neutrophil (NEU) count, C-reactive protein (CRP), aspartate aminotransferase, alanine aminotransferase (ALT), lactate dehydrogenase, urea levels in serum, fibrinogen, and hs-CRP levels [7]. Conversely, these patients demonstrated lower levels of serum albumin, white blood cells (WBC), procalcitonin (PCT), leukocytes, lymphopenia, and eosinopenia [8,9,10]. The study conducted by Gao et al. (2020) revealed significant differences between mild and severe COVID-19 cases in terms of interleukin-6 (IL-6), d-dimer (d-D), glucose, fibrinogen, thrombin time, and C-reactive protein levels [11].

Imaging also plays a significant role in the diagnosis of COVID-19. Computed tomography (CT) is strongly recommended for the diagnosis of COVID-19 [12] and plays an indispensable role in predicting, managing, and diagnosing COVID-19 pneumonia. CT severity scores have further been proposed as a useful tool for predicting disease severity and prognosis [13]. The CT severity score predicts COVID-19 disease severity by showing the score of lung involvement. It is considered the most effective method for quantitative assessment of the severity of COVID-19 pneumonia [13,14,15]. The CT severity score or index is defined as a scoring system that is used to assess lung involvement and changes caused by COVID-19 pulmonary infection [16]. The scoring system is based on an approximate estimation of pulmonary involvement. There are a few different ways to calculate the CT severity score. One commonly used scoring system is the 20-point scoring system, which assigns points based on the percentage of lung involvement and the presence of certain CT findings [16]. In this system, each lung is divided into five lobes, and each of the five lung lobes has been given a score from 1 to 5 based on a visual scoring [16], where number 1 means less than 5% lobar involvement, number 2 means 5–25% lobar involvement, number 3 represents 26–50% lobar involvement, number 4 means 51–75% lobar involvement, and finally, number 5 means more than 75% lobar involvement. All individual lobar scores are summed together to give a final score with a total score of 25 [17]. Another scoring system is the 25-point scoring system, which is similar to the 20-point system but includes additional points for the presence of pleural effusion or lymphadenopathy [16]. According to [18], a score of 7 or less (out of 25) is considered mild severity, while from 8–17 is moderate, and above 18 is considered severe. Other variations in scoring systems may include the use of different cutoffs for the percentage of lung involvement or the inclusion of other CT findings. Regardless of the specific scoring system used, the CT severity score index provides a quantitative measurement of lung involvement in COVID-19 patients, which can be used to assess disease severity and guide treatment decisions.

The CT severity score has been found to be positively correlated with laboratory parameters such as C-reactive protein (CRP), lactate dehydrogenase (LDH), and D-dimer, and with clinical parameters such as oxygen saturation, respiratory rate, and the need for mechanical ventilation [18,19]. The CT severity score was higher in patients with severe COVID-19 pneumonia compared to those with mild or moderate disease. However, the relationship between CT severity score and laboratory findings, and clinical parameters in COVID-19 pneumonia is not well understood. This systematic review aims to investigate this relationship and provide insights into the clinical utility of CT severity scores in COVID-19 pneumonia. Understanding the connection between chest CT severity score, laboratory findings, and clinical outcomes in COVID-19 pneumonia is crucial as the pandemic continues to spread around the world. We hypothesize that the CT severity score index is positively correlated with laboratory findings and negatively correlated with clinical outcomes, and that CT severity scores are higher in severe cases than in mild or moderate cases.

## 2. Methods

We used the recommendations from the Preferred Reporting Items for Systematic Reviews and Meta-analyses (PRISMA) guidelines to provide high-quality reporting [20]. We also used the PRISMA flow chart for the selection process (see Figure 1).

### 2.1. Search Strategy

We used three online databases, including Embase Ovid (1974–20 May 2023), MEDLINE Ovid (1946–20 May 2023), and PubMed (Inception–20 May 2023), to identify studies reporting the relationship between chest CT severity score and laboratory findings and clinical parameters in COVID-19 according to the PRISMA. We used the following search terms, including: “novel coronavirus” OR “SARS-CoV-2” OR “COVID-19” AND “Chest CT Severity Scores” OR “Severity Score” OR “Chest Severity Scoring System” AND “Clinical Parameters” OR “Laboratory Findings”. The literature found to be relevant was saved for further review to extract important information. The search was restricted to articles published in English. Studies were evaluated for compliance with the inclusion and exclusion criteria outlined below.

### 2.2. Selection Criteria

Three authors (NM, AA, and TR) independently screened the title, abstract, and full text of identified articles and assessed their eligibility for inclusion in this systematic review. Studies were included if they satisfied all inclusion criteria, and the referenced papers of included studies were evaluated for eligibility. Only published articles were considered for inclusion. The full texts of selected studies were downloaded for in-depth reading and information extraction in accordance with the primary purpose and objective of this study.

### 2.3. Inclusion and Exclusion Criteria

In this systematic review, because we intend to analyze the chest imaging appearances of patients with COVID-19, only studies meeting the following criteria were included: (a) the study included only COVID-19 patients; (b) original research studies reporting the association between chest CT severity score and laboratory findings and/or clinical characteristics in COVID-19 patients; (c) studies published in a journal and in English; (d) the study included a quantitative measure of chest CT severity score; (e) studies published on or after 2 March 2020; (f) studies reporting only CT but not X-ray severity score; (g) the study included laboratory findings such as white blood cell count, lymphocyte count, C-reactive protein (CRP), and procalcitonin (PCT); and (h) the study reported clinical outcomes such as mortality, ICU admission, and length of hospital stay. We excluded the following studies: (a) studies that did not report original data or clear diagnostic criteria, and (b) editorials, commentaries, opinions, all types of reviews, case reports, letters, and conference abstracts.

### 2.4. Data Extraction

Three authors (NM, AK, and NA) independently screened the titles and abstracts of the identified articles. Full-text articles were then reviewed to determine eligibility for inclusion. Data were extracted from the selected studies, including the basic characteristics of the included literature such as author, publication date, research type, number of patients/participants, study design, and methods (CT parameters). We also extracted the basic characteristics of the research subjects, such as age (mean age), sex, clinical manifestations, laboratory test results, and CT severity score.

### 2.5. Quality Assessment

Two authors (NM and WA) conducted the quality assessments of the studies independently using the critical appraisal checklist (checklist for analytical cross-sectional studies) suggested by the Joanna Briggs Institute (JBI) (https://jbi.global/critical-appraisal-tools) (accessed on 24 June 2023). Any discrepancies were resolved through consensus. The checklist comprises a set of eight questions that the authors considered for every study. Each question was assigned one point for an affirmative response. Hence, the ultimate scores assigned to each study have the potential to vary between 0 and 8. The scoring of each domain is categorized as either low risk, high risk, or unclear risk of bias. The risk of bias figure was created by Excel. Based on the overall score, we classified studies as low (≥7), moderate (4–6), or high risk of bias (≤1–3).

## 3. Results

### 3.1. Study Selection

A total of 80 citations were obtained through electronic searches of MEDLINE Ovid, Embase Ovid, and PubMed. After eliminating the duplicates, 43 studies remained for screening. When these remaining studies were screened, 22 studies were excluded. Two studies were excluded because they were abstracts and/or letters to the editor. Seven studies were also excluded because they did not investigate the association between the CT severity score and laboratory and clinical parameters. Three studies were further excluded because they were duplicates. Finally, ten studies were excluded due to not being free (no full text available). As a result, a total of 20 studies were considered for this analysis. Detailed steps for selecting studies are shown in Figure 1 (the PRISMA chart).

### 3.2. Characteristics of Studies

Appendix A summarizes the characteristics of the included studies (21 studies that were chosen for further examination). In brief, the studies in this review were conducted in various locations. Two studies [21,22] were conducted in Italy. Six studies [19,23,24,25,26,27] were conducted in India. Furthermore, two studies [18,28] were carried out in Pakistan. Two studies [29,30] were conducted in China. Three studies [31,32,33] were also carried out in Iran. One study [34] was conducted in Egypt. The four remaining studies were conducted in Belgium [35], Amsterdam, the Netherlands [36], Romania [37], and Martinique, France [38]. The studies reported on the following laboratory and clinical parameters: white blood cell (WBC) count, lymphocyte count, CRP, LDH, D-d, procalcitonin, ferritin, interleukin-6 (IL-6), arterial blood gas (ABG) analysis, and clinical severity scores. All studies were cross-sectional and retrospective, except one study [34], which was conducted prospectively. All studies were published within the timeframe of 2020–2023. The majority of studies (17 out of 20 studies) in this review used the common approach for calculating the CT severity score, which was based on the extent of lung involvement observed in chest CT scans using the semi-quantitative method or the so-called COVID-19 Reporting and Data System (CO-RADS) scoring system, ranging from 1 to 5, with higher scores indicating more severe disease involvement. Sample sizes ranged from 30 to 4004 COVID-19 patients, for a total of more than 8000 patients. The mean age of patients ranged from 18 to 90 years. Appendix A provides more characteristics and information on eligible studies on patients confirmed with COVID-19, including age, gender, CT severity score calculation, and main findings.

#### 3.2.1. Summary of Results Related to the Association between the CT Severity Score and Laboratory Findings

The current review evaluated the association of the CT severity score with laboratory findings and clinical outcomes and determined whether using the CT severity score can add extra value. The review revealed that five previous studies showed a positive association between the CT severity score and laboratory findings. Studies [25,29,30,39,40,41] have found that CT severity score was positively correlated with CRP, LDH, PCT, IL-6, D-dimer levels, blood glucose levels, blood sugar levels, neutrophils, and raised CRP levels. It has also been reported that COVID-19 patients with diabetes or hyperglycemia tended to have higher CT severity scores. In addition, a study [35] has attempted to find a relationship between vitamin D deficiency on admission and CT severity score, but this relationship was not significant. Furthermore, a previous study [37] found that a high baseline value for inflammatory biomarkers (such as monocyte-lymphocyte ratio (MLR), neutrophil-lymphocyte ratio (NLR), systemic immune-inflammation index (SII), and IL-6) and chest CT severity score was a strong predictor of adverse outcomes, including the need for invasive mechanical ventilation (IMV) and mortality, in patients with COVID-19 pneumonia. It has also been found that patients above 70 with atrial fibrillation (AF), dyslipidemia, and unvaccinated status highly predicted IMV need and fatality. In summary, all the studies concur that there is a correlation between the CT severity score and laboratory parameters.

#### 3.2.2. Summary of Results Related to the Association between the CT Severity Score and Clinical Parameters

The review also evaluated the correlation between the CT severity score and clinical parameters. Several studies showed a significant association between the CT severity score and clinical outcomes. Studies [22,23,24,25,26,28,33,34,38,40,41,42], found that the CT severity score is positively associated with mortality and morbidity rates (increased risk of mortality and morbidity), the need for mechanical ventilation and for emergency medical treatment, patient survival, inflammatory markers, endotracheal intubation, oxygen saturation and demand, respiratory rate, length of hospital stays (prolonged hospitalization, >21 days), increased risk of noninvasive positive-pressure ventilation (NIPPV) failure, ICU admission, vaccination status, age, gender, co-morbidities, and stage of disease in the cohort of COVID-19 patients. It has also been found that CT severity score is higher in patients with severe COVID-19 pneumonia compared to those with mild or moderate disease. Furthermore, it has been reported that (a) the mean CT severity score among COVID-19 patients who died was significantly higher than patients who survived; (b) COVID-19 patients with specific chest CT imaging features, such as ground glass opacities and consolidation, were more likely to have severe clinical outcomes; (c) COVID-19 patients with a higher CT severity score at admission had a higher risk of developing acute respiratory distress syndromes; and (d) fully vaccinated patients had a lower mean CTSS compared to unvaccinated or partially vaccinated patients, suggesting that full vaccination can aid in reducing the severity of lung involvement in COVID-19 infection. A reverse relationship between the CT severity score and blood oxygen saturation level has been demonstrated [34], which has major clinical implications, suggesting that, in patients with COVID-19 infection, clinicians should pay closer attention to the CT severity score. However, a previous study [21] has shown that the CT severity score did not predict in-hospital mortality in COVID-19 patients (it was not different between survivors and non-survivors). However, they found that only the SII on admission can independently predict in-hospital mortality in COVID-19 patients (after adjusting for confounders) and may assist with early risk stratification for this group.

### 3.3. Quality Assessment

Figure 2 and Table 1 below demonstrate the quality assessment (risk of bias) and quality scores for all studies. A total of 13 of the 20 assessed studies were of high quality (QS > 70–100%), 6 were of good quality (QS of 40–69%) and 2 studies [25,39] were of low quality (QS of 10–35%). There was no high risk of bias regarding the objective and standard criteria used for the measurement of the condition and statistical analyses. This was because the condition (CT severity score) was determined by a valid tool or subjectively by one or two radiologists, and appropriate statistical analyses were used. Overall, all studies were considered to have a low risk of bias in almost all domains, including inclusion criteria, description of subjects and setting, validity of exposure, measurement of condition, confounding factors, strategies for confounding factors, validity of outcome, and statistical analysis.

## 4. Discussion

The present systematic review has established a robust affirmative correlation between the CT severity score and laboratory as well as clinical parameters in patients affected by COVID-19. This suggests that the CT severity score can serve as a reliable prognostic indicator of the severity of the disease. The study suggests that there is a significant positive correlation between the severity score of CT scans and the levels of CRP, LDH, and D-dimer. This finding indicates that these markers may have potential usefulness in both the diagnosis and treatment of COVID-19 pneumonia. The observed positive correlation between the CT severity score and various clinical parameters such as oxygen saturation, respiratory rate, and the requirement for mechanical ventilation implies that the CT severity score can serve as a reliable indicator for identifying patients who may necessitate more intensive therapeutic interventions. The outcomes of this systematic review are in line with prior research that has assessed the correlation between chest CT severity score and laboratory results (including CRP, PCT, and IL-6 level) as well as clinical consequences (such as mortality and ICU admission) in COVID-19 pneumonia. Collectively, the results suggest that the CT severity score possesses the capability to function as a prognostic tool for predicting the severity of COVID-19 and guiding clinical decision making [43,44]. Furthermore, this approach could be deemed a dependable technique for evaluating the gravity of COVID-19 pneumonia. The utilization of this approach may have the capacity to facilitate risk stratification and short-term prediction for individuals with COVID-19, consequently reducing the burden on healthcare infrastructures amid the pandemic. Admitting patients with higher CT severity scores to the ICU unit at an early stage may offer advantages in reducing both mortality and morbidity. Importantly, from a clinical perspective, the utilization of the CT severity score system in the majority of the studies incorporated in this review is noteworthy. The scoring system, which is predicated on the degree and distribution of lung involvement on chest CT scans, has gained widespread adoption in several countries, including China, Italy, India, Iran, and certain regions in the Middle East [33,45,46,47,48]. Further investigation is necessary to corroborate these findings and establish a standardized CT scoring approach for COVID-19.

### Limitations

Although the present study offered a thorough examination of the existing literature demonstrating the role of CT severity score in predicting or diagnosing COVID-19 severity compared to laboratory and clinical parameters, it is important to acknowledge certain constraints that were present in this investigation. First, CT scans may not be readily available in all healthcare settings, and there are concerns about the potential harm from radiation exposure associated with repeated scans. Second, the CT severity score may not be specific to COVID-19 pneumonia and may be affected by other comorbidities. Third, despite conducting a comprehensive literature search, it is plausible that pertinent studies were overlooked and consequently excluded. Fourth, it is noteworthy that all the studies incorporated in this review were published articles, indicating the presence of publication bias that warrants attention. Furthermore, certain studies incorporated in this review exhibited relatively limited sample sizes, potentially diminishing the efficacy of certain conclusions drawn. The review suggests the use of the CT severity score in diagnosing COVID-19 or predicting its severity. However, this topic is subject to ongoing research and debate. The use of this scoring system in other countries as a diagnostic tool for COVID-19 pneumonia is still controversial and may be limited by differences in imaging protocols and interpretation. However, others have argued that the chest CT severity score may not be specific enough to differentiate COVID-19 from other respiratory diseases. The lack of standardization in the reporting of CT severity scores across studies is a potential limitation and may impact the generalizability of the findings. In comparison, laboratory findings and clinical parameters such as CRP, PCT, and oxygen saturation are widely available and less invasive measures that can be used to assess disease severity and guide management decisions in COVID-19 patients. These parameters can also be used to monitor the response to treatment and guide adjustments in therapy. Finally, different institutions or medical professionals may use slightly different scoring systems based on their own expertise or preferences. Additionally, because the scoring of CT scans can be subjective and somewhat reliant on interpretation by the reader, there may be some variability in scoring between different readers even when using the same system. Therefore, this highlights the need for standardization and consistency in the use of CT scoring systems in future studies. This study may have been improved with the inclusion of additional evaluations. The current study focuses on the evaluation of all studies that have examined the correlation between the CT severity score and laboratory or clinical parameters. However, the evaluation of a study based on individual patient data can provide more accurate and reliable findings compared to an overall evaluation of studies. This approach allows researchers to analyze patient-level characteristics, such as age, gender, comorbidities, and disease severity, which can have a significant impact on the study outcomes. By analyzing individual patient data, researchers can identify specific subgroups of patients who are more susceptible to severe disease and adverse outcomes. This information can be used to tailor treatment strategies and interventions to improve patient outcomes. Moreover, analyzing individual patient data can help researchers identify potential confounding factors that may influence study outcomes. By controlling for these factors, researchers can obtain more accurate estimates of the treatment effect and reduce bias in the study results. However, analyzing individual patient data can be time consuming and resource intensive, especially for large studies. It requires extensive data collection, management, and analysis, which can pose significant challenges for researchers. Therefore, the choice of approach may depend on the research question, available resources, and the feasibility of the study design.

## 5. Conclusions

This review found a strong correlation between CT severity score and clinical laboratory findings in COVID-19 pneumonia. Patients with severe COVID-19 pneumonia tend to have higher CT severity scores and abnormal laboratory findings compared to those with a milder disease. While the CT severity score may be a useful tool for predicting disease severity and prognosis in patients with COVID-19 pneumonia, it should be used in conjunction with laboratory findings and clinical parameters to guide management decisions. The use of a CT severity score should be balanced against the potential risks and limitations associated with repeated imaging studies. Ultimately, a multidisciplinary approach that considers all available clinical, laboratory, and radiological data is necessary for optimal management of COVID-19 patients. Future research should explore the use of chest CT severity scores in larger patient populations and across different healthcare settings worldwide.

## Figures and Tables

**Figure 1 diagnostics-13-02223-f001:**
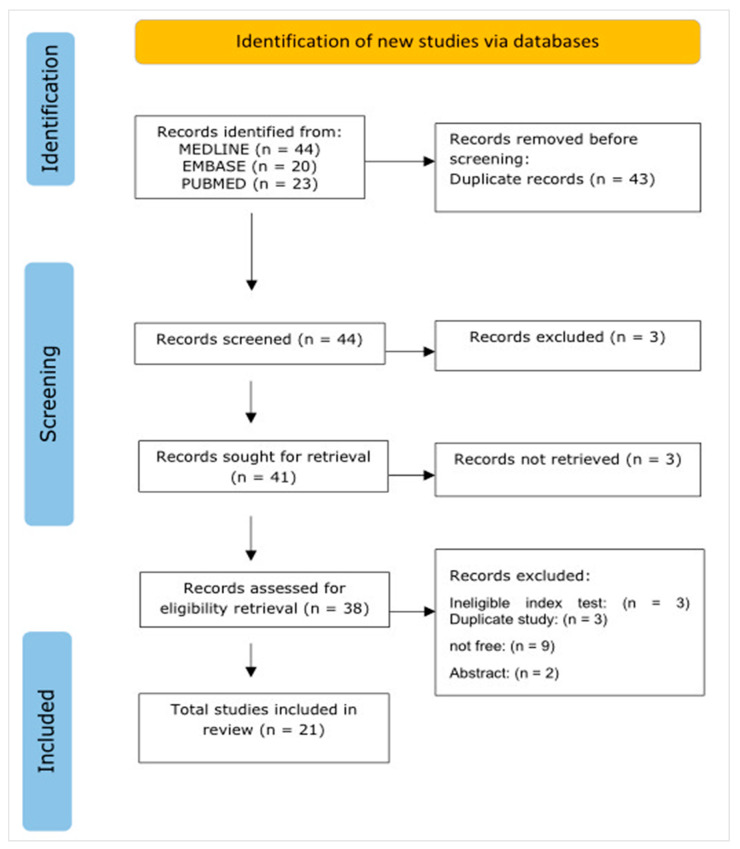
PRISMA flow chart for the selection process.

**Figure 2 diagnostics-13-02223-f002:**
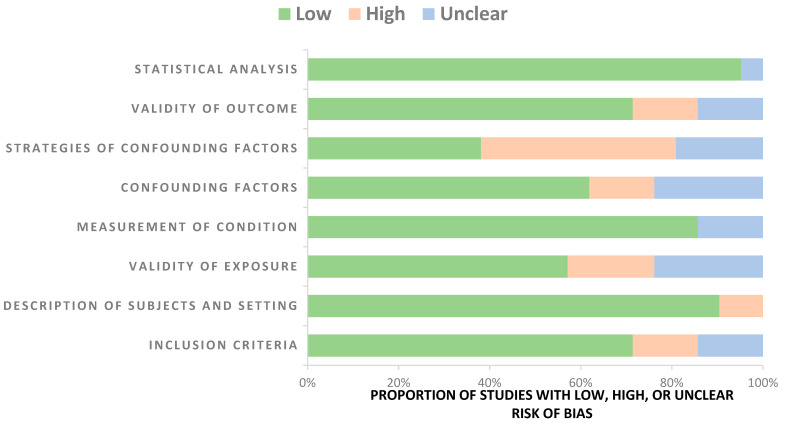
Risk of bias of the included studies with number of low, high, and unclear concerns in each study.

**Table 1 diagnostics-13-02223-t001:** Quality assessment of included studies using JBI tool for analytical cross-sectional studies.

Study	Inclusion Criteria	Description of Subjects and Setting	Validity of Exposure	Measurement of Condition	Confounding Factors	Strategies of Confounding Factors	Validity of Outcome	Statistical Analysis	Score (Out of 8)
Zinellu et al., 2021 [22]	☹	☺	?	?	☺	☹	☹	☺	3
Padmanaban et al., 2022 [25]	?	☹	?	☺	?	☹	?	☺	2
Fois et al., 2020 [21]	☹	☺	☺	☺	☺	☺	☺	☺	7
Younus et al., 2022 [28]	☺	☺	?	☺	?	☹	?	☺	4
Vishwanath et al., 2022 [27]	☺	☺	☺	☺	☺	☺	☺	☺	8
Lin et al., 2021 [29]	?	☺	☺	☺	?	?	☺	☺	5
Smet et al., 2020 [35]	?	☺	☺	☺	☺	☺	☺	☺	7
Yazdi et al., 2021 [41]	☺	☺	☺	☺	☺	☹	☺	☺	7
Abd El Megid et al., 2022 [34]	☺	☺	☺	☺	☺	☹	☺	☺	7
Arcari et al., 2022 [23]	☺	☺	☺	☺	?	?	☺	☺	6
Inamdar and Saboo, 2022 [19]	☹	☹	?	?	?	?	☺	?	1
Zhou et al., 2020 [30]	☺	☺	☹	☺	☺	☺	☺	☺	7
Valk et al., 2022 [36]	☺	☺	☹	☺	☺	☺	☺	☺	7
Abbasi et al., 2021 [31]	☺	☺	☹	☺	☺	☺	☺	☺	7
Halmaciu et al., 2022 [37]	☺	☺	☺	☺	☺	?	☺	☺	7
Atre et al., 2022 [24]	☺	☺	☺	☺	☺	☺	☺	☺	8
Aziz-Alhari et al., 2022 [33]	☺	☺	☺	☺	☹	☹	☺	☺	6
Saeed et al., 2021 [18]	☺	☺	?	?	☹	☹	☹	☺	3
Padelli et al., 2021 [38]	☺	☺	☺	☺	☺	☺	☺	☺	8
Sharma et al., 2022 [26]	☺	☺	☹	☺	☺	☹	☹	☺	5

Questions answered with yes, no, and unclear are given ☺, ☹,   ? symbols, respectively. The score was calculated as the sum of the questions answered with “yes”.

## Data Availability

Not applicable.

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
