# Peer review of "A Systematic Review of the Relationship between Chest CT Severity Score and Laboratory Findings and Clinical Parameters in COVID-19 Pneumonia"

_diagnostics, 2023, doi:10.3390/diagnostics13132223_

Round 1
Reviewer 1 Report
Overall comment on the abstract.
The author writes the aim of this article as "the relationship between CT severity score and clinical or laboratory parameters is not 19 yet well established. Here, we aim to provide a comprehensive systematic review of this relation". However, CT scan is expensive and difficult to conduct in many settings in comparison to other laboratory tests. The author writes "it may be feasible to forego the utilization of laboratory findings in this regard". However, there is no information on the quantitative analysis on the situation of forgoing the utilization of laboratory tests.
specific comments:
line 74-85 writes the CT scoring system but the supplementary document shows that the scoring system is not unique.
line 190-207 "Summary of results related to the association between CT severity score and laboratory findings". Whether all studies agreed on that the CT SS have relationship with laboratory parameters is not written.
Author Response
We would like to thank the reviewers for their thoughtful comments and suggestions. We have carefully addressed all required suggestions and in doing so feel the manuscript is substantially strengthened. We have also performed minor editing of the English language, as suggested. Attached are our detailed responses to their comments.
# Reviewer 1:
Overall comment on the abstract.
The author writes the aim of this article as "the relationship between CT severity score and clinical or laboratory parameters is not 19 yet well established. Here, we aim to provide a comprehensive systematic review of this relation". However, CT scan is expensive and difficult to conduct in many settings in comparison to other laboratory tests. The author writes "it may be feasible to forego the utilization of laboratory findings in this regard". However, there is no information on the quantitative analysis on the situation of forgoing the utilization of laboratory tests.
Response by the authors: Thank you for these fruitful comments, which undoubtedly improved the quality of our manuscript. Regarding the first point, although CT scan is expensive and not always feasible in certain settings, it remains a valuable tool in diagnosing and assessing the severity of certain medical conditions, especially when it comes to evaluating the severity of lung diseases such as COVID-19. The aim of this article is to provide a comprehensive review of the relationship between CT severity score and clinical or laboratory parameters that could potentially aid in clinical decision making and patient management, especially in healthcare settings where the CT scan is available. However, it is important to consider the limitations of CT scans, but also recognize their potential benefits in certain cases. We agree with the reviewer that the availability of CT scans may vary across healthcare settings and geographic regions. Consequently, we have revised the abstract to enhance its concision (please see page 1). We have also previously mentioned this point to be one of the limitations in our study. Please see the revised manuscript, page (9). Despite these limitations, the present study provides valuable insights into the potential role of CT severity score in predicting COVID-19 severity and its association with clinical and laboratory findings. It also highlights the need for further research in this area. This study may further be useful for healthcare professionals and researchers who have access to CT scan and are interested in understanding the relationship between CT severity score and clinical or laboratory parameters in patients with COVID-19.
Regarding the second point, the statement that "it may be feasible to forego the utilization of laboratory findings in this regard" suggests that laboratory tests may not be necessary in predicting or diagnosing COVID-19 severity, particularly if CT scans are available and indicate a clear severity score. However, the article does not provide a quantitative analysis of the situation regarding forgoing laboratory tests. Consequently, in accordance with the reviewer's suggestion, we have deleted this statement and revised the abstract (please see page 1).
specific comments:
line 74-85 writes the CT scoring system but the supplementary document shows that the scoring system is not unique.
Response by the authors: Thank you for this comment. Yes, we agree with the reviewer that the supplementary document shows different CT scoring systems. There are a few different ways to calculate the CT score, but the most commonly used method (a semi-quantitative scoring method previously used by Zhou et al. [8] and Wang et al. [4]) involves examining CT scans of the lungs and assigning a score for each lobe of the lung based on the amount of involvement. Each lobe can be assigned a score ranging from 0 (no involvement) to 5 (more than 75% involvement), with higher scores indicating more severe disease involvement. Additionally, a score can be assigned for the overall extent of lung involvement, ranging from 0 to 25. The scores for each lobe and the overall extent of involvement are then added together to obtain the total CT score. There are 17 studies (>80%) out of 21 using this scoring system or what is so called COVID-19 Reporting and Data System (CO-RADS) scoring system. Consequently, as the reviewer commented on this point, we have added a paragraph about the calculation of the CT severity score in the introduction section. We have also mentioned this point in the results section. Please see the revised manuscript, pages (2-3 in the introduction and page 6 in the results). In addition, due to different CT severity scoring systems, this highlights the need for standardization and consistency in the use of CT scoring systems in future studies. Thus, based on this, we have mentioned the different of CT severity scoring systems to be also one of the limitations in this study. Please see pages (9).
line 190-207 "Summary of results related to the association between CT severity score and laboratory findings". Whether all studies agreed on that the CT SS have relationship with laboratory parameters is not written.
Response by the authors: We have addressed the reviewer’s comment. What we have found from the previous studies is that there is a consensus on the relationship between CTSS and laboratory parameters. We have added a new sentence related to this point at the end of that paragraph. Please see the revised manuscript, page (6).

Reviewer 2 Report
Dear Editor, I found that manuscript is suitable for publication and the reviewer recommended its publication after revision.
The authors contributed the work entitled “A systematic review of the relationship between chest CT severity score and laboratory findings and clinical parameters in COVID-19 Pneumonia".
This review covers the knowledge in the field and expands the current understanding of the field but it suffers from some shortcomings.
The main concern of this manuscript is related to the results. It seems that the results of this research can be replaced with better tables and figures. It seems that if the evaluation of the study is based on the "patients" rather than just the overall evaluation of the studies, it can provide more correct findings.
Generally, the writing should be polished to improve the readability.
Author Response
Point-by-point response to reviewers’ comments
Journal/ID: Diagnostics-2445439
We would like to thank the reviewers for their thoughtful comments and suggestions. We have carefully addressed all required suggestions and in doing so feel the manuscript is substantially strengthened. We have also performed minor editing of English language, as suggested. Attached are our detailed responses to their comments.
Reviewer #2:
Dear Editor, I found that manuscript is suitable for publication and the reviewer recommended its publication after revision.
The authors contributed the work entitled “A systematic review of the relationship between chest CT severity score and laboratory findings and clinical parameters in COVID-19 Pneumonia".
This review covers the knowledge in the field and expands the current understanding of the field but it suffers from some shortcomings.
The main concern of this manuscript is related to the results. It seems that the results of this research can be replaced with better tables and figures. It seems that if the evaluation of the study is based on the "patients" rather than just the overall evaluation of the studies, it can provide more correct findings.
Response by the authors: Thank you for this comment. We have replaced the tables and figures in the results section, as suggested by the reviewer. Please see the results section in the revised manuscript, pages (7-10). Regarding the second comment, we agree with the reviewer’s suggestion that when the evaluation of the study is based on individual patients, rather than just the overall evaluation of the studies could provide more accurate and meaningful findings. However, the aim of this article is to systematically review the previous studies investigating the association between the CT severity score and laboratory or clinical findings in the diagnosis of COVID-19 pneumonia. Evaluating a study based on individual patient data can provide more accurate and reliable findings compared to an overall evaluation of studies. This approach can help identify patient subgroups that may benefit from specific interventions and improve patient outcomes. However, it also requires significant resources and careful consideration of the study design and analysis plan. In response to the reviewer's suggestion, we have pointed to the importance of this point in the discussion section. Please see page (12).
Comments on the Quality of English Language
Generally, the writing should be polished to improve the readability.
Response by the authors: Thank you for improving the quality of our manuscript. As suggested by the reviewer, we have made some minor language editing. Please see the entire document.
Thank you so much

Round 2
Reviewer 1 Report
I appreciate the corrections.
Reviewer 2 Report
_